# The NFLikelihood: An unsupervised DNNLikelihood from normalizing flows

**Humberto Reyes-Gonzalez**[1,2,3]⋆ **and Riccardo Torre**[2]†

**1** Department of Physics, University of Genova, Via Dodecaneso 33, 16146 Genova, Italy
**2** INFN, Sezione di Genova, Via Dodecaneso 33, I-16146 Genova, Italy
**3** Institut für Theoretische Teilchenphysik und Kosmologie,
RWTH Aachen, 52074 Aachen, Germany

⋆ humberto.reyes@rwth-aachen.de , † riccardo.torre@ge.infn.it

## Abstract

We propose the NFLikelihood, an unsupervised version, based on Normalizing Flows, of the DNNLikelihood proposed in Ref. [1]. We show, through realistic examples, how Autoregressive Flows, based on affine and rational quadratic spline bijectors, are able to learn complicated high-dimensional Likelihoods arising in High Energy Physics (HEP) analyses. We focus on a toy LHC analysis example already considered in the literature and on two Effective Field Theory fits of flavor and electroweak observables, whose samples have been obtained through the HEPFit code. We discuss advantages and disadvantages of the unsupervised approach with respect to the supervised one and discuss a possible interplay between the two.

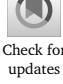

# 1 Introduction

The distribution, preservation, and reinterpretation of experimental and phenomenological Likelihoods arising in High Energy Physics (HEP) and astrophysics is an important and open topic [2]. In Ref. [1] it was shown how deep learning can play a crucial role in this context, by showing how the problem of encoding the Likelihood function into a Deep Neural Network (DNN) can be formulated as a supervised learning problem of regression. In simple terms, the values of the parameters $x$ and of the corresponding Likelihood $y = \mathcal{L}(x)$ are used to train a fully connected multilayer perceptron (MLP), which delivers a "pseudo-analytical" representation of the Likelihood function in terms of a DNN, therefore called DNNLikelihood.

In a recent paper, we have shown that Normalizing Flows (NFs) of the coupling and autoregressive type, are able to perform density estimation of very high dimensional probability density functions (PDFs) with great accuracy and with limited training samples and hyperparameters tuning [3]. Moreover, trained NFs, can be used as sample generators with two different approaches: on the one hand one can draw samples from the base distribution and transform them through the generative direction of the NFs, obtaining samples distributed according to the target PDF; on the other hand, the normalizing direction of the NFs can be used to get a fast prediction of the density for a given sample, allowing one to use the NF to assist and speed up traditional sequential Monte Carlo techniques [4–11]. Furthermore, NFs have been found useful to address a variety of other challenges in HEP such as event generation, unfolding, anomaly detection, etc. [12].

In this paper we show how Autoregressive Normalizing Flows (ANF) can be used to learn complicated Likelihoods, doing density estimation starting from the $x$ samples only and therefore offering an unsupervised approach to the DNNLikelihood.[1] We call the Likelihood encoded by NFs the NFLikelihood. The aim of this paper is twofold: on the one hand we want to propose the NFLikelihood as an alternative DNNLikelihood, discussing advantaged and disadvantages of the unsupervised approach, with respect to the supervised one. On the other hand we want to give explicit physics examples of the performances of the Autoregressive Flows studied in Ref. [3], which only considered toy distributions based on mixtures of Gaussians, by following a similar testing approach.

One important remark is that, since we are interested here in discussing the NF performances in learning some physical complicated densities, we focused on learning the posterior probability (not just the Likelihood), since these were the data we had at our disposal. Our approach can of course be trivially extended to the Likelihood by removing the contribution of the (known) prior used to sample the posterior.

The paper is organized as follows. In Section 2 we briefly describe the three phenomenological Likelihoods that we consider. Section 3 contains a discussion of the figures of merit that we used in our analysis, while Section 4 presents the main results. Finally, we report our conclusions in Section 5.

# 2 Likelihood functions for LHC analyses

In this analysis, we consider three Likelihoods of different dimensionality. We briefly describe them in turn in the following subsections.

---

[1]We are aware of the paper [13] proposing a similar approach in a different context.

## 2.1 The LHC-like new physics search Likelihood

As a first example we consider the toy LHC-like NP search, here after referred as the Toy Likelihood, introduced in Ref. [14] and also considered in Ref. [1]. We refer the reader to those references for a detailed explanation of the Likelihood construction, its parameters, and its sampling. Here we limit ourselves to remind that the Likelihood depends on one signal strength parameter $\mu$ and 94 nuisance parameters $\delta$.

## 2.2 The ElectroWeak fit Likelihood

The second Likelihood we consider is the one corresponding to the ElectroWeak fit presented in Ref. [15], which includes the recent top quark mass measurement by the CMS Collaboration [16] and $W$ boson mass measurement by the CDF Collaboration [17]. Such Likelihood, that we call EW Likelihood, depends on 40 parameters: 32 nuisance parameters and 8 parameters of interest, corresponding to the Wilson coefficients of the relevant Standard Model Effective Field Theory (SMEFT) operators. A sampling of the posterior probability distribution has been obtained with the HEPFit code [18]. The complete list of parameters with their definitions is reported in Appendix A.

In this case, the 1D marginal distributions of the parameters are all nearly Gaussian, with the exception of two truncated Gaussians, so that we expect it to be relatively simple for a NF with a Gaussian base distribution to learn the posterior. Nevertheless, the posterior shows strong correlations among some pairs of parameters (see Figure 5 in Appendix A), which helps to understand the ability of the NFs to accurately learn the correlation matrix.

## 2.3 The Flavor fit Likelihood

The third Likelihood we consider corresponds to the EFT fit to flavor observables related to neutral current $b \rightarrow s$ transitions presented in Ref. [19]. This Likelihood, referred to as the Flavor Likelihood, depends on 89 parameters: 77 nuisance parameters and 12 parameters of interest, corresponding to the Wilson coefficients of the relevant SMEFT operators. A sampling of the posterior probability distribution has been obtained with the HEPFit code documented in Ref. [18]. The full list of parameters is reported in Appendix A. This Likelihood is clearly more complicated than the previous two, since it features multimodal 1D distributions and complicated correlations (see Figs. 3 and 4).

# 3 Evaluation metrics

We used as quality metrics the mean over dimensions of the $p$-values of 1D Kolmogorov-Smirnov test (KS-test), with an optimal value of 0.5 and the Sliced Wasserstien distance (SWD) [20, 21], with optimal value 0. We briefly recall their definitions here for convenience:

- **Kolmogorov-Smirnov Test (KS)**
  The Kolmogorov-Smirnov (KS) test serves as a statistical test for assessing if two one-dimensional samples originate from the same underlying (unknown) probability density function (PDF). The null hypothesis assumes that both sets of samples are derived from the same PDF. The KS metric can be expressed as:

$$D_{y,z} = \sup_x |F_y(x) - F_z(x)|, \tag{1}$$

  where $F_{y,z}(x)$ is the empirical cumulative distribution functions of the sample sets $\{y_i\}$ and $\{z_i\}$, while sup denotes the supremum function. The $p$-value for null hypothesis

rejection is given by:

$$D_{y,z} > \sqrt{-\ln\left(\frac{p}{2}\right) \times \frac{1 + \frac{n_z}{n_y}}{2n_z}} \,, \tag{2}$$

where $n_y$ and $n_z$ indicate the sample sizes.

- **Sliced Wasserstein Distance (SWD)**
  The SWD serves as a metric for comparing two multi-dimensional distributions, leveraging the one-dimensional Wasserstein distance. The one-dimensional Wasserstein distance between two empirical distributions is formulated as:

$$W_{y,z} = \int_{\mathbb{R}} dx \, |F_y(x) - F_z(x)| \,. \tag{3}$$

In our sliced approach, we randomly select $N_d = 2D$ directions, with $D$ the dimensionality of the sample, uniformly distributed over the $4\pi$ solid angle.[2] We then project all samples on such directions and compute the one-dimensional Wasserstein distance and finally take the mean over the directions. For each SWD computation a new random selection of directions is drawn. See Ref. [3] for a more detailed discussion of the SWD in this context.

In order to include statistical uncertainty on the test and NF generated samples we compute the above metrics 100 times, for independent batches of $N_{\text{test}}/100$ points, and take the average.

With respect to the Ref. [3] we also consider here the metric given by the discrepancy on the Highest Posterior Density Interval (HPDI). This is a very important metric for Bayesian posterior inference, since it tells how well credibility intervals (CI) are reproduced by the NFLikelihood. In particular, we computed the HPDI relative error width (HPDIe) for 68.27%, 95.45%, and 99.73% (CI) of each 1D marginal of the true and predicted distributions. For each dimension, we compute the mean of this quantity when more than one interval is present (which is common for multimodal distributions). Finally, we take the median over all dimensions. We choose the median to avoid that results on very noisy dimensions, particularly in the Flavor Likelihood, have a large negative effect on the generally good value of the metric.

# 4   The NFLikelihood

The results of this analysis have been obtained using the TENSORFLOW2 NF implementation from Ref. [3]. All models were trained with Masked Autoregressive Flow (MAF) architectures [23]. The difference is the type of bijector used. For the Toy Likelihood, we employed affine bijectors as in the original MAF paper. Here on, we referred to this specific architecture as MAF. For the EW and Flavor Likelihoods we implemented the Rational Quadratic Spline bijector [24] structure. We denote the corresponding architecture as A-RQS. We always trained with a log-probability loss function. For all three cases the training data was standardized (to zero mean and unit standard deviation) before training and a small scan over the flow's hyper-parameters was performed. Here we only present the optimal results obtained for each distribution. All training iterations were performed with an initial learning rate of 0.001, reduced by a factor of 0.2 after a *patience* number of epochs without improvement on the validation loss. Training was early stopped after $2 \cdot patience$ number of epochs without improvement. The value of *patience* and of the other relevant hyper-parameters will be reported separately for each of the Likelihoods. All models have been trained on Tesla V100 Nvidia GPUs.

---

[2]This is achieved by normalizing an $N$-dimensional vector whose components are sampled from independent standard normal distributions [22].

Table 1: Hyperparameters leading to the best determination of the Toy Likelihood.

| Hyperparameters for Toy Likelihood | | | | | | | | |
|---|---|---|---|---|---|---|---|---|
| # of train samples | hidden layers | algorithm | # of bijec. | spline knots | range | L1 factor | patience | max # of epochs |
| $2 \cdot 10^5$ | $3 \times 64$ | MAF | 2 | - | - | 0 | 20 | 200 |

Table 2: Best results obtained for the Toy Likelihood.

| Results for Toy Likelihood | | | | | | |
|---|---|---|---|---|---|---|
| # of test samples | Mean KS-test | Mean SWD | $HPDIe_{1\sigma}$ | $HPDIe_{2\sigma}$ | $HPDIe_{3\sigma}$ | time (s) |
| $2 \cdot 10^5$ | $0.4893 \pm .0292$ | $0.03947 \pm .0019$ | 0.02073 | 0.01207 | 0.01623 | 133 |

Table 3: Results for the POI in the Toy Likelihood.

| Results for Toy Likelihood POI | | | |
|---|---|---|---|
| POI | KS-test | $HPDIe_{1\sigma}$ | $HPDIe_{2\sigma}$ | $HPDIe_{3\sigma}$ |
| $\mu$ | 0.54 | 0.02742 | 0.01359 | 0.01786 |

## 4.1 The Toy Likelihood

The hyperparameters that lead to the best estimation of the Toy Likelihood are shown in Table 1. The corresponding NF architecture is made of two MAF bijectors and one reverse permutation between them. Each MAF has an autoregressive network with 3 hidden layers made of 64 nodes each. The training was performed for a maximum of 200 epochs, with $patience = 20$ and $2 \cdot 10^5$ training samples. The NF model was tested with $2 \cdot 10^5$ test samples.

The resulting quality metrics are shown in Table 2. In particular, we obtained an optimal KS-test of $\sim 0.49$ and HPDIe of the order of $10^{-2}$, which guarantee that, within the considered statistical uncertainty, the NF generated samples are indistinguishable from those generated with the true pdf. The training time was about 133s. Since, when doing inference from a Likelihood function or posterior distribution, one is usually specially interested in the so-called parameters of interests (POIs), we show in Table 3 the results obtained for $\mu$. Here the KS-test is again $\sim 0.5$ and HPDIes of the order of $10^{-2}$. The accuracy of the NF model is visually shown in Figure 1, which presents a corner plot of a selection of 10 parameters, including $\mu$. In the Figure, the true distribution is shown in red, while the NF distribution in blue. The HPDIs corresponding to 68.27% ($1\sigma$), 95.45% ($2\sigma$), and 99.73% ($3\sigma$) probabilities are shown as solid, dashed and dashed-dotted lines, respectively. The selected parameters include those considered in Ref. [1], therefore allowing for a direct comparison. In particular, comparing with what in Ref. [1] is called the Bayesian DNNLikelihood, we find that both approaches gives extremely accurate results: the NF approach seems to perform slightly better. However, the main difference seems to be in the training time, which is much larger for the DNNLikelihood. The advantage of the DNNLikelihood with respect to the NFLikelihood comes when the so-called Frequentist Likelihood is considered: in this case one is not particularly interested in learning the Likelihood (or the posterior) as a PDF, but is instead interested in learning it as a function close to its absolute and local (profiled) maxima. This highlights the main difference between the DNNLikelihood and the NFLikelihood. The first is more suitable to encode Likelihoods to be used for frequentist analyses, while the second for Likelihoods (or posteriors) to be used in Bayesian analyses. Obviously one can combine the two approaches to obtain a general and flexible representation of the Likelihood suitable for both frequentist and Bayesian inference. We defer this generalization to future work.

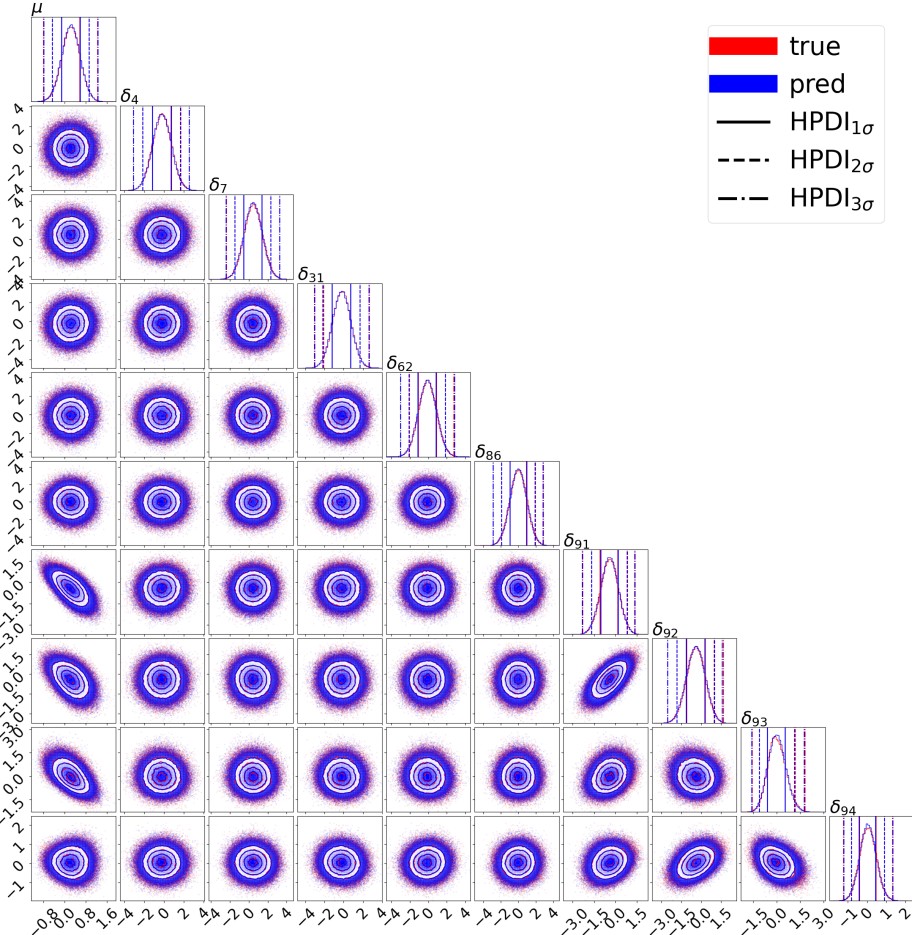

Figure 1: Corner plot of the 1D and 2D marginal posterior distributions of a representative selection of the Toy Likelihood parameters. The true distribution is depicted in red, while the predicted distribution is shown in blue. The solid, dashed and dashed-dotted line over the 1D marginals denote the $68.27\%, 95.45\%$, and $99.73\%$ HPDIs, respectively. The rings on the 2D marginals describe the corresponding probability levels.

## 4.2 The EW Likelihood

The hyperparameters corresponding the best NF model describing the EW Likelihood are shown in Table. 4. The chosen NF architecture is made of two A-RQS bijectors with 4 spline knots defined in a $[-6, 6]$ range, and one reverse permutation between them. Each A-RQS has an autoregressive network with 3 hidden layers made of 128 nodes each. The training was performed for a maximum of 800 epochs and a patience of 20 with $2 \cdot 10^5$ training samples. The NF model was tested with $2 \cdot 10^5$ samples. Finally, given the presence of truncated dimensions, the distributions was soft clipped, with a hinge factor of $10^{-4}$ at the truncations, within the range of the training data[3].

A summary of the values obtained for the evaluation metrics is reported in Table 5. We obtained a mean KS-test of $\sim 0.4$ and HPDIes of the order of $10^{-2}$ or smaller. The training time was about 7200s, that is a couple of hours. Furthermore, Table 6 shows the metrics obtained for the Wilson coefficients (POIs). We find that most of the POIs are pretty well described, albeit small discrepancies found for $C_{\varphi l}^1$, $C_{\varphi l}^3$ and $C_{ll}$ which can be likely fixed after fine-tuning

---

[3]This was done via the soft-clip bijector from TENSORFLOW-PROBABILITY [25]. The hinge factor was chosen to be $\lll 1$ to obtain an approximate identity mapping within the defined range. Note however that this may lead to numerically ill-conditioned boundaries if the discrepancy between distributions is significant.

Table 4: Hyperparameters leading to the best determination of the EW Likelihood.

| Hyperparameters for the EW Likelihood | | | | | | | | |
|---|---|---|---|---|---|---|---|---|
| # of train samples | hidden layers | # of bijec. | algorithm | spline knots | range | L1 factor | patience | # of epochs |
| $2 \cdot 10^5$ | 2 | $3 \times 128$ | A-RQS | 4 | -6 | 0 | 20 | 800 |

Table 5: Best results obtained on the EW Likelihood.

| Results for the EW Likelihood | | | | | | |
|---|---|---|---|---|---|---|
| # of test samples | Mean KS-test | Mean SWD | $\text{HPDIe}_{1\sigma}$ | $\text{HPDIe}_{2\sigma}$ | $\text{HPDIe}_{3\sigma}$ | time (s) |
| $2 \cdot 10^5$ | $0.4307 \pm 0.06848$ | $0.003131 \pm 0.00053$ | 0.000339 | 0.0008664 | 0.006973 | 7255 |

Table 6: Results for the Wilson coefficients in the EW Likelihood.

| Results for EW Likelihood | | | | |
|---|---|---|---|---|
| POI | KS-test | $\text{HPDIe}_{1\sigma}$ | $\text{HPDIe}_{2\sigma}$ | $\text{HPDIe}_{3\sigma}$ |
| $c_{\varphi l}^1$ | 0.1901 | 0.08384 | 0.09787 | 0.437 |
| $c_{\varphi l}^3$ | 0.2078 | 0.0346 | 0.1039 | 0.4967 |
| $c_{\varphi q}^1$ | 0.4581 | 0.02279 | 0.01131 | 0.04866 |
| $c_{\varphi q}^3$ | 0.4989 | 0.01219 | 0.01439 | 4.1017 |
| $c_{\varphi d}$ | 0.5221 | 0.01713 | 0.03808 | 0.09952 |
| $c_{\varphi e}$ | 0.4885 | 0.01453 | 0.2146 | 0.1401 |
| $c_{\varphi u}$ | 0.5259 | 0.005409 | 0.005082 | 0.341 |
| $c_{ll}$ | 0.2193 | 0.1667 | 0.08047 | 0.0713 |

the hyper-parameters, increasing the number of trainable parameters or adding more training points.

The true and NF distributions are visually compared in Figure 2, which shows a corner plot over the POIs plus four representative nuisance parameters (a total of twelve parameters). As before, the distribution is represented in red, while the NF distribution in blue. The HPDIs corresponding to 68.27%, 95.45%, and 99.73% probabilities are shown as solid, dashed and dashed-dotted lines, respectively. We see that in general, the NF distributions matches pretty well the true one. Something worth emphasizing is the NF ability to learn even large correlations between dimensions. This is not expected in the case of the DNNLikelihood, since regression becomes inefficient when large correlations between parameters are present.

## 4.3 Flavor Likelihood

The optimal hyperparameters found for learning the Flavor Likelihood are shown in Table 7. The chosen NF architecture is made of two A-RQS bijectors with 8 spline knots defined in the $[-5, 5]$ range, and one reverse permutation between them. Each A-RQS has an autoregressive network with 3 hidden layers made of 1024 nodes each and an L1 regularization factor of $10^{-4}$. The training was performed for a maximum of 12000 epochs with a patience of 50. The model was trained with $10^6$ samples and tested with $5 \cdot 10^5$ samples. Furthermore, since the Likelihood function presents several truncated dimensions, the NF model was soft-clipped, with an hinge factor of $1 \cdot 10^{-4}$, within the range of the training data.

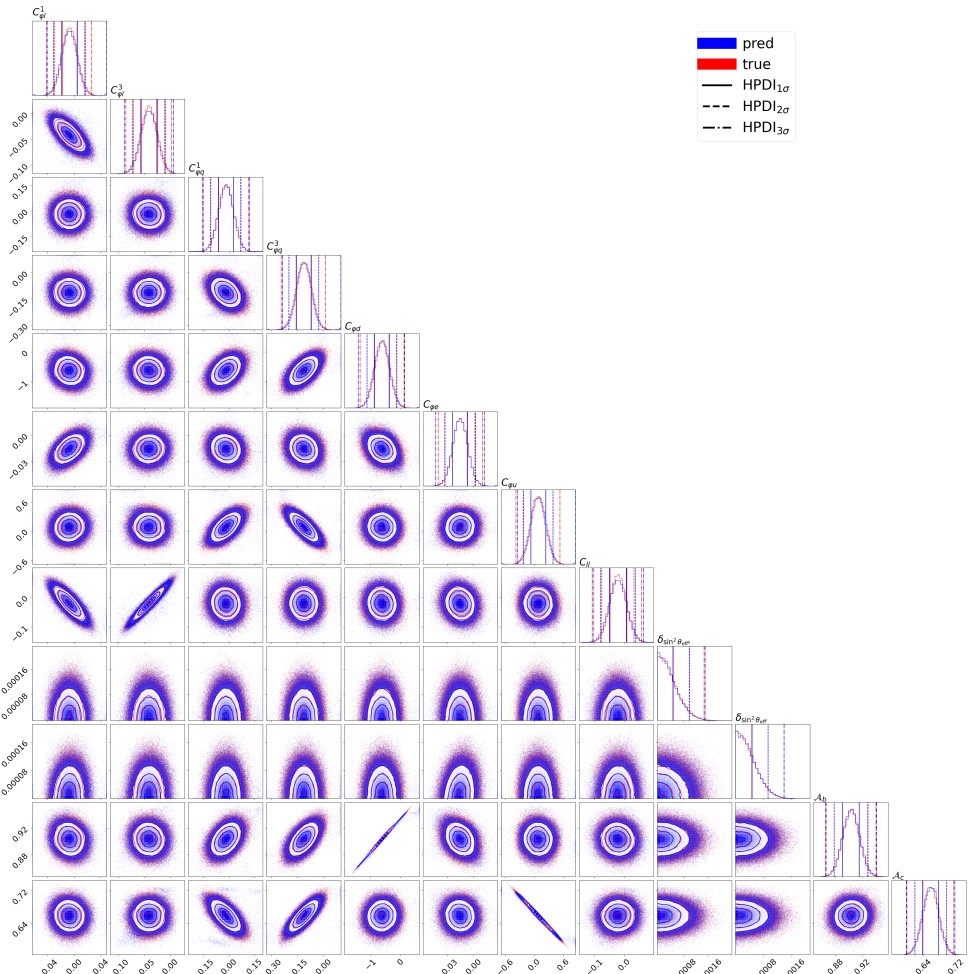

Figure 2: Corner plot of the 1D and 2D marginal posterior distributions of the POIs plus four representative nuisance parameters of the EW Likelihood. The true distribution is depicted in red, while the predicted distribution is shown in blue. The solid, dashed and dashed-dotted line over the 1D marginals denote the 68.27%, 95.45%, and 99.73% HPDIs, respectively. The rings on the 2D marginals describe the corresponding probability levels.

A summary of the evaluation metrics is shown in Table 8. We obtained a good KS-test of $\sim 0.42$ and HPDIes of the order of $10^{-2}$ or smaller. Training took about $1 \cdot 10^4$s, i.e. around 2.7 hours. The Flavor Likelihood includes 12 Wilson coefficients as POIs, and Table 9 shows the results obtained for each of them. The majority of the KS-tests are above 0.4, with a few exceptions where the value is still above 0.3. In turn, the HPDIes are generally of the order $10^{-2}$ or smaller, with some exceptions, that we believe may be improved by again finely-tuning the architecture, increasing the number of trainable parameters or by adding more training points.

It is important to stress the complexity of the Flavor likelihood. As can be seen from Figures 3 and 4, depicting a corner plot of the Wilson coefficients and the 1D marginal distributions of all dimensions, respectively, the posterior features multimodal 1D marginals, complex correlations and noisy dimensions, offering a very realistic prototype of a complicated high dimensional HEP Likelihood. Nonetheless, we find that the NF model is able to reproduce it with a very good accuracy.

Table 7: Hyperparameters leading to the best determination of the Flavor Likelihood.

| Hyperparameters for the Flavor Likelihood | | | | | | | | |
|---|---|---|---|---|---|---|---|---|
| # of train samples | hidden layers | # of bijec. | algorithm | spline knots | range | L1 factor | patience | max # of epochs |
| $10^6$ | $3 \times 1024$ | 2 | A-RQS | 8 | -5 | 1e-4 | 50 | 12000 |

Table 8: Best results obtained for the Flavor Likelihood.

| Results for the Flavor Likelihood | | | | | | |
|---|---|---|---|---|---|---|
| # of test samples | Mean KS-test | Mean SWD | $\text{HPDIe}_{1\sigma}$ | $\text{HPDIe}_{2\sigma}$ | $\text{HPDIe}_{3\sigma}$ | time (s) |
| $5 \cdot 10^5$ | $0.4237 \pm 0.03405$ | $0.02717 \pm 0.002374$ | 0.00867 | 0.007346 | 1.419e-07 | 9550 |

Table 9: Results for the Wilson coefficients in the Flavor Likelihood.

| Results for Flavor Likelihood POIs | | | | |
|---|---|---|---|---|
| POI | KS-test | $\text{HPDIe}_{1\sigma}$ | $\text{HPDIe}_{2\sigma}$ | $\text{HPDIe}_{3\sigma}$ |
| $c_{1123}^{LQ1}$ | 0.4346 | 0.007251 | 1.83e-05 | 4.731e-08 |
| $c_{2223}^{LQ1}$ | 0.4736 | 0.01249 | 0.00162 | 0.03575 |
| $c_{1123}^{Ld}$ | 0.486 | 0.01466 | 0.006628 | 0.002338 |
| $c_{2223}^{Ld}$ | 0.4138 | 0.0513 | 0.02446 | 2.398e-08 |
| $c_{11}^{LedQ}$ | 0.5362 | 0.00738 | 0.004683 | 5.387e-08 |
| $c_{22}^{LedQ}$ | 0.5161 | 0.02799 | 0.001639 | 2.155e-09 |
| $c_{2311}^{Qe}$ | 0.4476 | 0.01389 | 0.007458 | 1.419e-07 |
| $c_{2322}^{Qe}$ | 0.382 | 0.02132 | 0.02496 | 0.0004609 |
| $c_{1123}^{ed}$ | 0.4789 | 0.04076 | 0.00333 | 5.602e-08 |
| $c_{2223}^{ed}$ | 0.4436 | 0.008685 | 0.016 | 1.502e-08 |
| $c_{11}'^{LedQ}$ | 0.3203 | 0.09194 | 0.007041 | 8.011e-08 |
| $c_{22}'^{LedQ}$ | 0.4157 | 0.03001 | 0.008749 | 4.374e-08 |

# 5  Conclusion

The publication of full Likelihoods is crucial for the long lasting legacy of the LHC, and for any other experiment involving complicated analyses with a large parameter space. However, this is not always a straightforward matter since Likelihoods are often high dimensional complex distributions, sometimes depending on Monte Carlo simulations and/or numeric integrations, which make their sampling a very hard task. Furthermore, one requires precise, compact, and efficient representations of them so that they can be easily and systematically reused. As it was first shown in Ref. [1], Neural Networks, being universal interpolators, offer a promising approach to encode, preserve, and reuse Likelihood functions. In this work we extended this approach to unsupervised learning, proposing the use of Normalizing Flows for this endeavour. Indeed, Normalizing Flows are powerful generative models which, by construction, also provide density estimation. We tested our proposal on three posterior distributions of increasing complexity, corresponding to three different Likelihood functions: a 95-dimensional LHC-like new physics search Likelihood, a 40-dimensional ElectroWeak EFT fit Likelihood, and an 89-

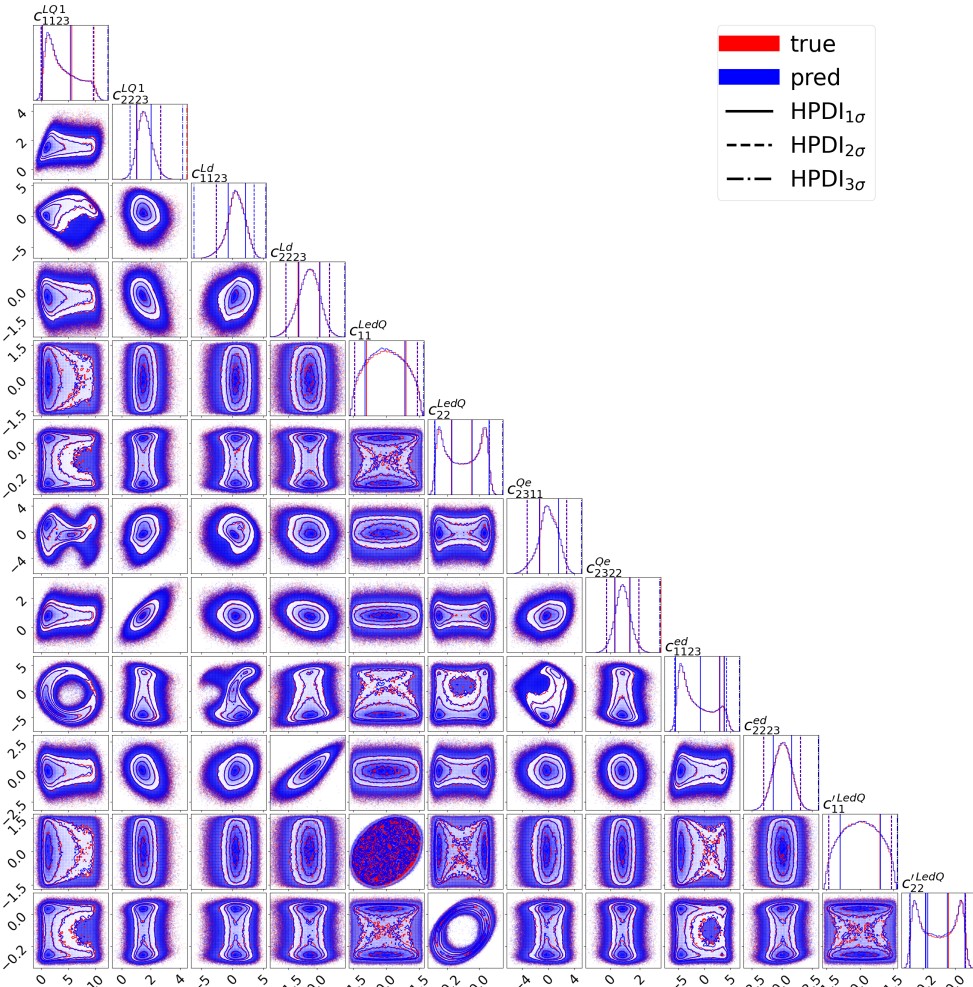

Figure 3: Corner plot of the 1D and 2D marginal posterior distributions of the Wilson coefficients of the Flavor Likelihood. The true distribution is depicted in red, while the predicted distribution is shown in blue. The solid, dashed and dashed-dotted line over the 1D marginals denote the 68.27%, 95.45%, and 99.73% HPDIs, respectively. The rings on the 2D marginals describe the corresponding probability levels.

dimensional Flavor EFT fit Likelihood. We found that Autoregresive Normalizing Flows are capable of precisely describing all the above examples, including all the multimodalities, truncations, and complicated correlations. In fact, we see that, given the way they are constructed, Autoregressive Flows can easily learn the covariance matrices of the distributions. Both the code used for this project [26] and a user-friendly TENSORFLOW2 framework for Normalizing Flows (still under development) [27] are available on GitHub. The training and generated data, as well as the trained NF models, are available on Zenodo [28].

The 95-dimensional LHC-like new physics search Likelihood, which was also studied in the context of the DNNLikelihood of Ref. [1] was also used to make a comparison between the two approaches. Such comparison leads to the conclusion that the two approaches are complementary and could, in the future, be merged to get an even more flexible representation of the Likelihood. Indeed, while the DNNLikelihood approach focuses on learning the Likelihood as a multivariate function, and is agnostic about its probability interpretation, the NF approach leverages the latter. This implies that the DNNLikelihood approach is more suitable to learn Likelihood functions to be used in Frequentist analyses, where the region of profiled maxima is

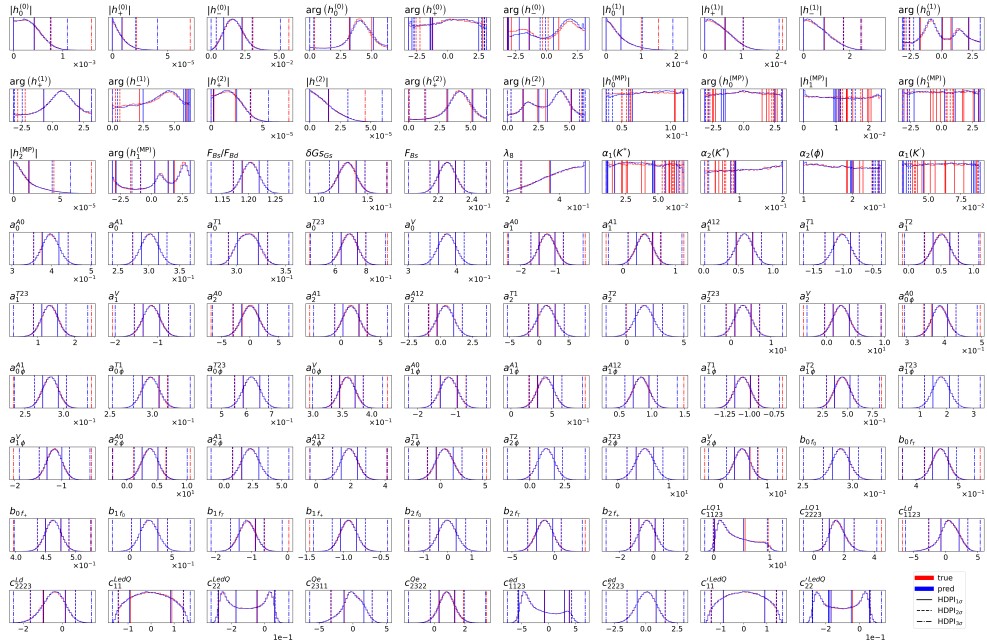

Figure 4: 1D marginal posterior distributions of all the parameters of the Flavor Likelihood. The true distribution is depicted in red, while the predicted distribution is shown in blue. The solid, dashed and dashed-dotted lines over the marginals denote the 68.27%, 95.45%, and 99.73% HPDIs, respectively.

the important part to learn, while the NFLikelihood approach is more suitable for Likelihoods (or posteriors) to be used in Bayesian analyses, where is crucial to learn the distribution as a statistical PDF and not just a multivariate function. We defer to future work the study of the best approach to merge the DNN and NF Likelihoods into a unique object.

As a follow-up, we also plan to explore the possibility of learning full statistical models, i.e. functions of both the data and the parameters. A promising way to do this is by means of the so-called conditional Normalizing Flows [29].

# Acknowledgements

We thank Luca Silvestrini for useful discussions and for providing the samples of the EW and Flavor Likelihoods. We also thank the IT service of INFN Sezione di Genova, and especially Mirko Corosu, for computing support. H.R.G. is also thankful to Sabine Kraml, Wolfgang Waltenberger, and Danny van Dyk for encouraging discussions.

**Funding information**  This work was supported by the Italian PRIN grant 20172LNEEZ. HRG also acknowledges support from the Deutsche Forschungsgemeinschaft (DFG, German Re- search Foundation) under grant 396021762 – TRR 257: Particle Physics Phenomenology after the Higgs Discovery.

# A  Details of the EW and Flavor Likelihoods

The list of parameters and their description for the EW Likelihood is given in Table 10, while Figure 5 gives a pictorial representation of the data correlation matrix. The list of parameters and their description for the Flavor Likelihood is given in Table 11.

Table 10: Parameters of the EW Likelihood.

| # | Parameter | Description | # | Parameter | Description |
|---|---|---|---|---|---|
| 1 | $\alpha_S(M_Z)$ | SM input | 21 | $C_{\varphi u}$ | Wilson coefficient |
| 2 | $\Delta\alpha_{\text{had}}^{(5)}(M_Z)$ | SM input | 22 | $C_{ll}$ | Wilson coefficient |
| 3 | $M_Z$ | SM input | 23 | $P_\tau^{\text{pol}}$ | Observable |
| 4 | $m_H$ | SM input | 24 | $M_W$ | Observable |
| 5 | $m_t$ | SM input | 25 | $\Gamma_W$ | Observable |
| 6 | $\delta_{\Gamma_Z}$ | SM input uncertainty | 26 | $\text{BR}_{W\to l\bar{\nu}_l}$ | Observable |
| 7 | $\delta_{M_W}$ | SM input uncertainty | 27 | $\mathcal{A}_s$ | Observable |
| 8 | $\delta_{R_b^0}$ | SM input uncertainty | 28 | $R_{uc}$ | Observable |
| 9 | $\delta_{R_c^0}$ | SM input uncertainty | 29 | $\sin^2\theta_{\text{eff}}$ | Observable |
| 10 | $\delta_{R_l^0}$ | SM input uncertainty | 30 | $\Gamma_Z$ | Observable |
| 11 | $\delta_{\sin^2\theta_{\text{eff}^b}}$ | SM input uncertainty | 31 | $\sigma_h^0$ | Observable |
| 12 | $\delta_{\sin^2\theta_{\text{eff}^l}}$ | SM input uncertainty | 32 | $R_l^0$ | Observable |
| 13 | $\delta_{\sin^2\theta_{\text{eff}^q}}$ | SM input uncertainty | 33 | $A_{\text{FB}}^{0,l}$ | Observable |
| 14 | $\delta_{\sigma_h^0}$ | SM input uncertainty | 34 | $\mathcal{A}_l$ | Observable |
| 15 | $C_{\varphi l}^1$ | Wilson coefficient | 35 | $R_b^0$ | Observable |
| 16 | $C_{\varphi l}^3$ | Wilson coefficient | 36 | $R_c^0$ | Observable |
| 17 | $C_{\varphi q}^1$ | Wilson coefficient | 37 | $A_{\text{FB}}^{0,b}$ | Observable |
| 18 | $C_{\varphi q}^3$ | Wilson coefficient | 38 | $A_{\text{FB}}^{0,c}$ | Observable |
| 19 | $C_{\varphi d}$ | Wilson coefficient | 39 | $\mathcal{A}_b$ | Observable |
| 20 | $C_{\varphi e}$ | Wilson coefficient | 40 | $\mathcal{A}_c$ | Observable |

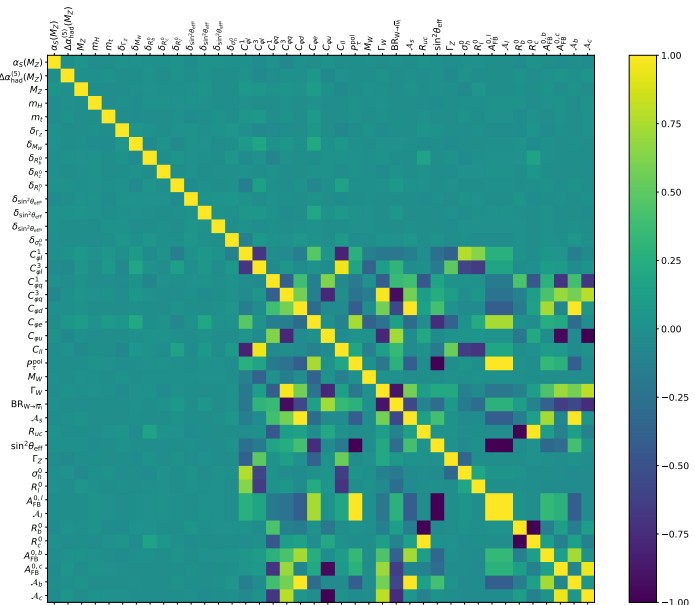

Figure 5:  Correlation matrix of the ElectroWeak fit data.

Table 11: Parameters of the Flavor Likelihood.

| # | Parameter | Description | # | Parameter | Description | # | Parameter | Description |
|---|---|---|---|---|---|---|---|---|
| 1 | $\lvert h_0^{(0)} \rvert$ | Nuis | 31 | $a_0^{A0}$ | Nuis | 61 | $a_{1\phi}^{V}$ | Nuis |
| 2 | $\lvert h_+^{(0)} \rvert$ | Nuis | 32 | $a_0^{A1}$ | Nuis | 62 | $a_{2\phi}^{A0}$ | Nuis |
| 3 | $\lvert h_-^{(0)} \rvert$ | Nuis | 33 | $a_0^{T1}$ | Nuis | 63 | $a_{2\phi}^{A1}$ | Nuis |
| 4 | $\arg(h_0^{(0)})$ | Nuis | 34 | $a_0^{T23}$ | Nuis | 64 | $a_{2\phi}^{A12}$ | Nuis |
| 5 | $\arg(h_+^{(0)})$ | Nuis | 35 | $a_0^{V}$ | Nuis | 65 | $a_{2\phi}^{T1}$ | Nuis |
| 6 | $\arg(h_-^{(0)})$ | Nuis | 36 | $a_1^{A0}$ | Nuis | 66 | $a_{2\phi}^{T2}$ | Nuis |
| 7 | $\lvert h_0^{(1)} \rvert$ | Nuis | 37 | $a_1^{A1}$ | Nuis | 67 | $a_{2\phi}^{T23}$ | Nuis |
| 8 | $\lvert h_+^{(1)} \rvert$ | Nuis | 38 | $a_1^{A12}$ | Nuis | 68 | $a_{2\phi}^{V}$ | Nuis |
| 9 | $\lvert h_-^{(1)} \rvert$ | Nuis | 39 | $a_1^{T1}$ | Nuis | 69 | $b_{0f_0}$ | Nuis |
| 10 | $\arg(h_0^{(1)})$ | Nuis | 40 | $a_1^{T2}$ | Nuis | 70 | $b_{0f_T}$ | Nuis |
| 11 | $\arg(h_+^{(1)})$ | Nuis | 41 | $a_1^{T23}$ | Nuis | 71 | $b_{0f_+}$ | Nuis |
| 12 | $\arg(h_-^{(1)})$ | Nuis | 42 | $a_1^{V}$ | Nuis | 72 | $b_{1f_0}$ | Nuis |
| 13 | $\lvert h_+^{(2)} \rvert$ | Nuis | 43 | $a_2^{A0}$ | Nuis | 73 | $b_{1f_T}$ | Nuis |
| 14 | $\lvert h_-^{(2)} \rvert$ | Nuis | 44 | $a_2^{A1}$ | Nuis | 74 | $b_{1f_+}$ | Nuis |
| 15 | $\arg(h_+^{(2)})$ | Nuis | 45 | $a_2^{A12}$ | Nuis | 75 | $b_{2f_0}$ | Nuis |
| 16 | $\arg(h_-^{(2)})$ | Nuis | 46 | $a_2^{T1}$ | Nuis | 76 | $b_{2f_T}$ | Nuis |
| 17 | $\lvert h_0^{(MP)} \rvert$ | Nuis | 47 | $a_2^{T2}$ | Nuis | 77 | $b_{2f_+}$ | Nuis |
| 18 | $\arg(h_0^{(MP)})$ | Nuis | 48 | $a_2^{T23}$ | Nuis | 78 | $c_{1123}^{LQ1}$ | Wilson coefficient |
| 19 | $\lvert h_1^{(MP)} \rvert$ | Nuis | 49 | $a_2^{V}$ | Nuis | 79 | $c_{2223}^{LQ1}$ | Wilson coefficient |
| 20 | $\arg(h_1^{(MP)})$ | Nuis | 50 | $a_{0\phi}^{A0}$ | Nuis | 80 | $c_{1123}^{Ld}$ | Wilson coefficient |
| 21 | $\lvert h_2^{(MP)} \rvert$ | Nuis | 51 | $a_{0\phi}^{A1}$ | Nuis | 81 | $c_{2223}^{Ld}$ | Wilson coefficient |
| 22 | $\arg(h_1^{(MP)})$ | Nuis | 52 | $a_{0\phi}^{T1}$ | Nuis | 82 | $c_{11}^{LedQ}$ | Wilson coefficient |
| 23 | $F_{B_s}/F_{B_d}$ | Nuis | 53 | $a_{0\phi}^{T23}$ | Nuis | 83 | $c_{22}^{LedQ}$ | Wilson coefficient |
| 24 | $\delta Gs_{Gs}$ | Nuis | 54 | $a_{0\phi}^{V}$ | Nuis | 84 | $c_{2311}^{Qe}$ | Wilson coefficient |
| 25 | $F_{B_s}$ | Nuis | 55 | $a_{1\phi}^{A0}$ | Nuis | 85 | $c_{2322}^{Qe}$ | Wilson coefficient |
| 26 | $\lambda_B$ | Nuis | 56 | $a_{1\phi}^{A1}$ | Nuis | 86 | $c_{1123}^{ed}$ | Wilson coefficient |
| 27 | $\alpha_1(K^*)$ | Nuis | 57 | $a_{1\phi}^{A12}$ | Nuis | 87 | $c_{2223}^{ed}$ | Wilson coefficient |
| 28 | $\alpha_2(K^*)$ | Nuis | 58 | $a_{1\phi}^{T1}$ | Nuis | 88 | $c_{11}^{\prime LedQ}$ | Wilson coefficient |
| 29 | $\alpha_2(\phi)$ | Nuis | 59 | $a_{1\phi}^{T2}$ | Nuis | 89 | $c_{22}^{\prime LedQ}$ | Wilson coefficient |
| 30 | $\alpha_1(K^{\prime})$ | Nuis | 60 | $a_{1\phi}^{T23}$ | Nuis | | | |

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
