# Peer review of "The NFLikelihood: an unsupervised DNNLikelihood from Normalizing Flows"

_SciPost Physics, doi:SciPost Phys. Core 7, 048 (2024)_

## Round 1 · Referee Report · Anonymous (Referee 1) · 2024-1-24

Strengths

- towards ML implementation of LHC likelihoods: very much needed and method looks very promising

- chosen examples are high-dimensional and seem realistic

Weaknesses

- choice of evaluation metrics might not be capturing all correlations.

- one paragraph contradicts the tables it describes

- presentation is not good: lots of typos and other issues

Report

Review of the scipost submission 2309.09743v1, entitled "The NFLikelihood: an unsupervised DNNLikelihood from Normalizing Flows", by Humberto Reyes-Gonzalez and Riccardo Torre.
The authors describe the use of normalizing flows, a type of deep generative network with an explicit likelihood, to encode and preserve high-dimensional experimental likelihoods for high-energy physics analyses. They consider 3 different likelihoods (a toy, an electroweak, and a flavor example) and show how well the model learns the distribution of the likelihood, using different evaluation metrics. To learn the likelihoods, the authors use a set of sampled points in parameter space that they obtained from other sources (like the HEPfit code).
The efficient encoding of experimental likelihoods is an important problem and normalizing flows seem a promising solution to it. I therefore think that the manuscript should be published in SciPost physics. However, there are a few things, mostly regarding the presentation of the results, that need to be improved before.

main points:

- The suggested evaluation metrics seem to only look at lower dimensional projections and not the entire distribution with all its correlations. The KS test looks at one-dimensional marginals. For the SWD, it is not clear to me how large the subspace is. The text says 2D directions, but a sample is only D-dimensional. Do the authors mean 2 directions?

- Instead of metrics that only evaluate subspaces, the authors could consider a classifier-based approach, as was discussed in 2305.16774, to ensure that all correlations are learned correctly.

- The authors say that they evaluated all metrics 100 times to include statistical uncertainty, yet all tables only report a central value and no errorbar. Please add the error so the numbers can be judged better.

- Tables 3, 6, and 9: What does it mean to compute the statistical tests for parameters of interest? Do they sample from the entire (joint) distribution of POI+nuissances and look at the one-dimensional distribution of the POI(s) only to compute the numbers? Please explain that better.

- Please give a formula for soft clipping and explain what a hinge factor is.

- When describing Table 6, I'm not sure "generally well described" is the best way to express when 3 of the KS-tests are of order 0.2.

- The paragraph explaining tables 8 and 9 contradicts the tables. The KS-test is 0.42 in the text and 0.31 in the table. The HPDIes are of order 10^-3 in the text and 10^-2 in the table. Training time is 20000s in the text and 9550s in the table. I strongly disagree with the statement "almost always above 0.4" when the KS-test of the POIs is described: only 4(!) of 12 POIs are above 0.4.

- Could the authors please be more specific why the large value for HPDIe_1sigma for c'_22^LedQ is due to the algorithm?

minor points:

- in the introduction, the authors state that "The aim of this paper is (...) to give explicit physics examples of the performances of the autoregressive flows studied in (3)". I think this sentence is misleading, it suggests that this was not done before. However, autoregressive flows have been used a lot in HEP on various physics datasets (event generation, detector simulation, anomaly detection, unfolding, ...).

- A notational question: when referring to the "MAF", the authors mean an autoregressive architecture with affine transformations and when they say "A-RQS" they mean the same architecture, but now with a spline-based transformation? Other HEP papers used "MAF" just to indicate the autoregressive nature and not what kind of transformation is used, which might be confusing to readers.

typos:

- section 2.3: refereed -> referred

- section 2.3: Fig.s 3 and 4 -> Figs. 3 and 4

- Table 1: column headlines of "# of bijections" and "algorithm" should be interchanged to match the body of the table.

- Table 3: Please check the caption, I think "flavor" should be example 3, not 1.

- Figure 1: There is a space missing right before the beginning of the second sentence in the caption.

- Section 4.2: Table.4 -> Table 4.

- Section 4.2: Last line of page 6: Table 3 -> Table 6.

- Table 4: column headlines of "# of bijections" and "hidden layers" should be interchanged to match the body of the table.

- Acknowledgements: Luce -> Luca

Requested changes

- see report for content

- please correct the typos I listed and check if there are more

- in general: please make all numbers/text in figures as big as the font of the main body of the paper. Labels in Figure 1 are borderline small, labels in Figures 2 and 4 are way too small.

  • validity: high
  • significance: high
  • originality: high
  • clarity: good
  • formatting: acceptable
  • grammar: good

Author:  Humberto Reyes-González  on 2024-04-10  [id 4405]

(in reply to Report 1 on 2024-01-24)

We thank the Referee for their careful reading and constructive suggestions. Below we reply separately to each of their comments.
**Referee**

*The authors describe the use of normalizing flows, a type of deep generative network with an explicit likelihood, to encode and preserve high-dimensional experimental likelihoods for high-energy physics analyses. They consider 3 different likelihoods (a toy, an electroweak, and a flavor example) and show how well the model learns the distribution of the likelihood, using different evaluation metrics. To learn the likelihoods, the authors use a set of sampled points in parameter space that they obtained from other sources (like the HEPfit code).The efficient encoding of experimental likelihoods is an important problem and normalizing flows seem a promising solution to it. I therefore think that the manuscript should be published in SciPost physics. However, there are a few things, mostly regarding the presentation of the results, that need to be improved before. *

**Main points:**

**Referee**
*The suggested evaluation metrics seem to only look at lower dimensional projections and not the entire distribution with all its correlations. The KS test looks at one-dimensional marginals. For the SWD, it is not clear to me how large the subspace is. The text says 2D directions, but a sample is only D-dimensional. Do the authors mean 2 directions?*
**Reply**
Our implementation of the sliced-Wasserstein distance (SWD), discussed at more length in our previous paper of Ref. [3], consists in the average of the $1D$ Wasserstein distances computed between a number (that we choose larger than the data dimensionality $D$, and equal to $2D$) of randomly generated projections, obtained by sampling directions uniformly on the surface of a unit $D$-dimensional sphere. We have added explicit reference to [3] at the end of the discussion of the SWD in the manuscript. Therefore, to answer to the question of the referee, the underlying Wasserstein distance calculation is only $1D$, but the average is taken over $2D$ projections. Notice that the number of projections is not limited to the number of dimensions. Indeed, differently from the case of the mean-KS metric, we do not consider just the $D$ projections along the "axes", which are just the $1D$ marginals, but we consider random $1D$ projections in the $D$-dimensional space. It is known that (see Refs. [20, 21] in the paper), under specific hypotheses, the average computed by the SWD approximates the true multidimensional Wasserstein distance (which is prohibitively expensive to compute) as the number of projections increases (and asymptotically approaches infinity). Therefore, despite being intrinsically $1D$, the SWD is able to (at least partially) capture the effect of correlations, while still remaining very efficient to compute. We would also like to add that the design and evaluation of metrics for high-dimensional distributions is an open problem, to which we are actively working and on which we have a forthcoming paper appearing shortly.

**Referee**
*Instead of metrics that only evaluate subspaces, the authors could consider a classifier-based approach, as was discussed in 2305.16774, to ensure that all correlations are learned correctly.*
**Reply**
We are aware of the powerful approach of using classifiers to evaluate the performance of generative models. Nevertheless, their applicability and interpretation in high dimensions is not straightforward. Again, in a forthcoming publication, we aim at comparing the performance of average-based metrics (like those used in this paper) with the classifier-based approach. Given the tangential nature of this issue with respect to the present work, and its intrinsic relevance in the field of generative models, we prefer to defer the discussion to a dedicated publication and use here simple, efficient, and interpretable metrics.

**Referee**
*The authors say that they evaluated all metrics 100 times to include statistical uncertainty, yet all tables only report a central value and no errorbar. Please add the error so the numbers can be judged better.*
**Reply**
Done for the overall KS and SWD metrics. For each individual dimension, the distribution is approximately uniform random, as expected.

**Referee**
*Tables 3, 6, and 9: What does it mean to compute the statistical tests for parameters of interest? Do they sample from the entire (joint) distribution of POI+nuissances and look at the one-dimensional distribution of the POI(s) only to compute the numbers? Please explain that better.*
**Reply**
That is correct.

**Referee**
*Please give a formula for soft clipping and explain what a hinge factor is.*
**Reply* *
We used the soft clip bijector from tensor flow-probability. In the manuscript, we provide the reference to TensorFlow Probability SoftClip (https://www.tensorflow.org/probability/api_docs/python/tfp/bijectors/SoftClip) in the manuscript. As described in the reference, the soft-clip is obtained by using the softplus bijector (https://www.tensorflow.org/probability/api_docs/python/tfp/bijectors/Softplus) as a smooth approximation of $\max(x,0)$. The soft-limits are obtained as: $$\begin{array}{lll} \max(x, \text{low}) & = \max(x - \text{low}, 0) + \text{low} \\ & \approx \text{softplus}(x - \text{low}) + \text{low} \\ & := \text{softlower}(x)\\ \min(x, \text{high}) & = \min(x - \text{high}, 0) + \text{high} \\ & = -\max(\text{high} - x, 0) + \text{high} \\ & \approx -\text{softplus}(\text{high} - x) + \text{high} \\ & := \text{softupper}(x) \end{array}$$ The soft plus, defined as $f_c(x) := c \cdot g(x / c) = c \cdot \log[1 + \exp(x / c)]$, where c is the hinge factor, maps the distribution to the domain of positive real numbers. The hinge parameter in the soft plus changes the transition at zero. In the soft-clip, $c<<1$ will approximate the identity mapping, but may be numerically ill-conditioned at the boundaries. If $c > 1.0$ translates to a smoother mapping, but creates a larger distortions. We chose $c<<1$ to avoid such distortions, since the distributions are well approximated at the tails, the error at the boundaries is generally small.

**Referee**
*When describing Table 6, I'm not sure "generally well described" is the best way to express when 3 of the KS-tests are of order 0.2.*
**Reply**
This has been rephrased to 'We find that most of the POIs are pretty well described, albeit small discrepancies found for $C_{\varphi l}^{1}$, $C_{\varphi l}^{3}$ and $C_{ll}$'

**Referee**
*The paragraph explaining tables 8 and 9 contradicts the tables. The KS-test is 0.42 in the text and 0.31 in the table. The HPDIes are of order $10^{-3}$ in the text and $10^{-2}$ in the table. Training time is 20000s in the text and 9550s in the table. I strongly disagree with the statement \"almost always above 0.4\" when the KS-test of the POIs is described: only 4(!) of 12 POIs are above 0.4.*
**Reply**
The discrepancies have been corrected.

**Referee**
*Could the authors please be more specific why the large value for HPDIe$_{1\sigma}$ for $c_{22}^{\prime LedQ}$ is due to the algorithm?*
**Reply**
The results have been updated. The original HPDIe$_{1\sigma}$ for $c_{22}^{\prime LedQ}$ did not correspond to the figure.
*Minor points:*

**Referee**
*in the introduction, the authors state that \"The aim of this paper is (\...) to give explicit physics examples of the performances of the autoregressive flows studied in (3)\". I think this sentence is misleading, it suggests that this was not done before. However, autoregressive flows have been used a lot in HEP on various physics datasets (event generation, detector simulation, anomaly detection, unfolding, \...).*
**Reply**
The variety of NF applications in HEP has been addressed. Our specific aim was to follow a similar testing strategy as in (3), while using the same implementations of the NFs.

**Referee**
*A notational question: when referring to the "MAF", the authors mean an autoregressive architecture with affine transformations and when they say "A-RQS" they mean the same architecture, but now with a spline-based transformation? Other HEP papers used "MAF" just to indicate the autoregressive nature and not what kind of transformation is used, which might be confusing to readers.*
**Reply**
Thanks for pointing this out. The naming conventions have been clarified in the manuscript.

**Referee**
*Please correct the typos I listed and check if there are more.*
**Reply**
We have corrected the typos and checked for more.

**Referee**
*In general: please make all numbers/text in figures as big as the font of the main body of the paper. Labels in Figure 1 are borderline small, labels in Figures 2 and 4 are way too small.*
**Reply**
*We have done our best to improve readability. We admit that some of the figures are not meant to be read on paper, but rather to be zoomed in on a screen. We have saved the figures in vector format with this purpose in mind.*

---

## Round 2 · Referee Report · Anonymous (Referee 1) · 2024-4-24

Strengths

  • towards ML implementation of LHC likelihoods: very much needed and method looks very promising

  • chosen examples are high-dimensional and seem realistic

  • much improved presentation compared to v1

Weaknesses

-

Report

The authors have addressed almost all points to my satisfaction. I have only 3 minor comments that can be addressed without showing me again:

  • When evaluating the SWD 100 times, are the 2D directions sampled anew in each of these 100 evaluations? Or is the same set of 2D directions used 100 times? Please add this info.

  • In order to improve the performance on the not-so-well fitted parameters, the authors write these "be likely fixed after further fine-tunning the hyper-parameters or adding more training points". I suggest adding "or adding more bijectors to the flow" as another option to increase expressivity.

  • There is one typo remaining that I had already pointed out: On p7, in section 4.2, "table 3" should be "table 6" in the text.

Requested changes

see report

Recommendation

Publish (easily meets expectations and criteria for this Journal; among top 50%)

  • validity: high
  • significance: high
  • originality: high
  • clarity: high
  • formatting: excellent
  • grammar: excellent

Author:  Humberto Reyes-González  on 2024-05-23  [id 4507]

(in reply to Report 1 on 2024-04-24)

We thank again the Referee for their careful reading and constructive suggestions. Below we reply separately to each of their comments.

Referee - When evaluating the SWD 100 times, are the 2D directions sampled anew in each of these 100 evaluations? Or is the same set of 2D directions used 100 times? Please add this info.

Response The 2D directions are randomly drawn each time independently. We clarify this in the manuscript.

Referee - In order to improve the performance on the not-so-well fitted parameters, the authors write these "be likely fixed after further fine-tunning the hyper-parameters or adding more training points". I suggest adding "or adding more bijectors to the flow" as another option to increase expressivity.

Response

Thank you for the suggestion. In previous studies with generic multi-modal distributions, we found that, at least for A-RQS, using more than 2 bijectors didn't make much difference as compared to enlarging the NNs and the number of knots. Of course, is hard to generalise this statement for all possible cases. To keep it general we now write: 'be likely fixed after further fine-tunning the hyper-parameters, increasing the number of trainable parameters or adding more training points.'

Referee - There is one typo remaining that I had already pointed out: On p7, in section 4.2, "table 3" should be "table 6" in the text.

Response Apologies for overlooking this. The typo has been corrected now.

---

## Round 2 · List of Changes

• Minor corrections and changes with respect to v1. -Minor changes to tables. -Updated results (marginal differences) including uncertainty estimation . -Added reference to the Soft-clip bijector. -Clarification about the naming conventions used for each Normalizing Flow architecture.

---

## Round 3 · Referee Report · Anonymous (Referee 2) · 2024-6-7

Strengths

1- Application of well-studied machine learning (ML) strategy to the novel problem of likelihood estimation (strictly speaking, posterior estimation)

2- Case studies on two realistic examples at the Large Hadron Collider (LHC).

Weaknesses

1- No real innovation on the ML side, just a direct application of normalizing flows.

2- Ultimate version of this method is likely to use conditional normalizing flows, which is not explored in this paper

Report

I was asked by the Editor to evaluate this manuscript in the context of suitability for SciPost Physics versus SciPost Physics Core.

First, let me concur with the previous referee that this paper presents a promising new approach towards an ML implementation of likelihoods relevant for the LHC. In previous work by one of the authors on DNNLikelihood, it was shown that one can use ML regression strategies to interpolate the likelihood function from a set of sampled points taken from the true likelihood. Here, the authors introduce NFLikelihood, which uses normalizing flows to learn the likelihood function (strictly speaking the posterior for a given prior). The authors discuss the relative strengths of the two approaches, where DNNLikelihood is typically better suited for frequentist analyses where the peaks of the likelihood are most important to model, while NFLikelihood is typically better suited for Bayesian analyses, where it is important to model the entire posterior. In addition to using a toy LHC analysis to benchmark against DNNLikelihood, the authors apply NFLikelihood to two realistic studies related to electroweak and flavor physics. These are high dimensional problems (the smallest example is 40 parameters), so the strong performance of NFLikelihood is very promising, especially given its reduced computational costs.

At some level, all this paper is doing is using normalizing flows for density estimation, which is a well studied problem. That said, this paper is performing density estimation in a context of high relevance to the LHC, where archiving likelihoods in a computational efficient form is an important part of making sure LHC analyses can be reused in the context of global fits. If this paper only repeated the toy LHC analysis from the DNNLikelihood paper, I would probably recommend publication in SciPost Physics Core. But since the authors both introduce a novel machine learning strategy and apply it to two realistic problems of interest to the LHC community, I think this paper meets the standards of SciPost Physics.

Requested changes

None

Recommendation

Publish (meets expectations and criteria for this Journal)

---

## Round 3 · List of Changes

-Minor clarification about Sliced Wasserstein Distance computation procedure.
-Minor clarification about further hyper parameter tuning.
- Fixed typo Table 3 -> Table 6.

---

## Editorial Decision

published